# Density-Based Unsupervised Learning Algorithm to Categorize College Students into Dropout Risk Levels

**Miguel Angel Valles-Coral \***, **Luis Salazar-Ramírez**, **Richard Injante**, **Edwin Augusto Hernandez-Torres**, **Juan Juárez-Díaz**, **Jorge Raul Navarro-Cabrera**, **Lloy Pinedo** and **Pierre Vidaurre-Rojas**

Facultad de Ingeniería de Sistemas e Informática, Universidad Nacional de San Martín, Jr. Maynas, Tarapoto 22200, Peru
* Correspondence: mavalles@unsm.edu.pe

**Abstract:** Compliance with the basic conditions of quality in higher education implies the design of strategies to reduce student dropout, and Information and Communication Technologies (ICT) in the educational field have allowed directing, reinforcing, and consolidating the process of professional academic training. We propose an academic and emotional tracking model that uses data mining and machine learning to group university students according to their level of dropout risk. We worked with 670 students from a Peruvian public university, applied 5 valid and reliable psychological assessment questionnaires to them using a chatbot-based system, and then classified them using 3 density-based unsupervised learning algorithms, DBSCAN, K-Means, and HDBSCAN. The results showed that HDBSCAN was the most robust option, obtaining better validity levels in two of the three internal indices evaluated, where the performance of the Silhouette index was 0.6823, the performance of the Davies–Bouldin index was 0.6563, and the performance of the Calinski–Harabasz index was 369.6459. The best number of clusters produced by the internal indices was five. For the validation of external indices, with answers from mental health professionals, we obtained a high level of precision in the *F*-measure: 90.9%, purity: 94.5%, *V*-measure: 86.9%, and ARI: 86.5%, and this indicates the robustness of the proposed model that allows us to categorize university students into five levels according to the risk of dropping out.

**Keywords:** clustering; data mining; DBSCAN; K-Means; HDBSCAN

## 1. Introduction

The university, within society, is the institution dedicated to teaching, research, and generation of new knowledge, where the student is the nucleus on which its main purposes and principles are governed [1]. Therefore, it is essential to have strategies and mechanisms that ensure the care and permanence of the student, as well as compliance with the basic conditions of educational quality that guarantee the adequate teaching–learning process [2].

During the life of the student, the university stage represents perhaps the most important challenge [3]; in general, this stage takes place at the end of secondary education, a moment when the student experiences big changes that range from the social to the emotional aspects [4]. Likewise, at that time, the student is exposed to new experiences and responsibilities that require high physical and mental effort [5].

This wears them down and makes them self-demanding, a combination that generates anxiety, a normal and involuntary response that produces physical and psychological symptoms [6]. Therefore, not meeting the psychological needs of the student generates problems that affect the learning process, their social development, and puts their health and life at risk [7]. In addition, the low or null attention to the problems of the students, by those responsible, generates dissatisfaction and low motivation for the development of academic activities and increases the chances of abandoning studies partially or totally [8,9].

Thus, university tutoring has turned out to be the fundamental support mechanism for monitoring the student's training process [10,11]. In this sense, universities must respond

to the challenges of providing comprehensive education, institutionalizing methods and procedures that guarantee the identification of students with academic risks and establishing preventive and corrective intervention programs to mitigate the probability of desertion [12]. Then, we can affirm that within the university, tutoring directs fundamental processes related to the attention and psychological follow-up of the student to promote professional development and projection [13].

Based on this, one of the topics of wide interest in university institutions is the need to have mechanisms and tools that help to face the phenomena related to the risk of student desertion [14]. Likewise, these tools must provide alert systems or strategies that allow us to intervene in the most vulnerable groups that have a higher level of risk of deserting [15]. Thus, we emphasize the use of ICT in the educational field [16–18], and automatic learning methods become the most viable option, since thanks to their characteristics they allow us to develop useful models capable of analyzing and discovering complex patterns in datasets. This allows us to model information for decision-making in the diagnosis and treatment of possible psychological interventions [19–21].

Consequently, our objective and main contribution is to propose an academic and emotional monitoring model that uses data mining and machine learning to group university students, according to their level of dropout risk. In this solution, we integrate ICT advances; specifically, chatbots for data collection and density-based unsupervised algorithms for student clustering, which serve as a basis for future projects and a precedent for other work and joint efforts between mental health professionals and ICT management.

The article is organized as follows: In Section 2, we describe the theoretical foundations of the variables under study, in Section 3 we present the experimental design, as well as the materials and methods we used, and in Section 4 we detail the analysis of the results found and the discussions. Finally, in Section 5, we presented conclusions and future implications of the research.

## 2. Theoretical Fundament

### 2.1. College Dropout

The phenomenon of university dropout is one of the main problems affecting educational systems [22], which is why it has been studied by different approaches, such as psychological, sociological, and economic [23]. Each approach has independently exposed the different perspectives and perceptions of the students regarding the main variables that motivated their attempt or action to dropout, encompassing them in two blocks, the academic aspect, and the individual (personal) aspect [24,25].

In the case of the academic aspect, the main variables related to desertion are the previous performance of the student (the knowledge that they have formed before entering the university, which can be used to develop their academic activities), emotional intelligence (ability to understand, use, and manage feelings and emotions appropriately), motivation, and individual learning objectives (the awareness of the knowledge acquired by the student, who puts it into practice in daily life).

For the individual aspect, the main variables are linked to the age and sex of the individual, their socioeconomic status, their social and interpersonal relationships, their mood, and their behavioral aspects [26–31].

These variables generate data and information of each student, and thanks to the advances of ICT in educational issues, mechanisms have been generated that apply data mining and machine learning, and data can be worked on and manipulated, regardless of whether they are static or dynamic. This facilitates analysis and control, and reflected in terms of precision, these mechanisms are very effective and require less effort for data processing, compared to the conventional methods used by mental health professionals [32–34].

Several studies use data mining techniques to find common and specific denominator patterns in student populations and group them to predict academic performance and the possibility of dropping out of academic activities, and based on this, generate imme-

diate solutions that mitigate the cases found, minimizing the dropout rate of university students [35–37].

*2.2. Density-Based Clustering*

Clustering based on point cloud density is an unsupervised learning methodology whose function is to identify specific groups in the data, based on the fact that a cluster is a region within a contiguous data space of high density of elements, dissociated from other similar clusters by contiguous zones of low density [38,39]. For a better definition, there are different approaches to classify what characterizes different groups in the data.

- Procedurally, the various clustering methods attempt to partition the data into *k* clusters, such that we minimize within-cluster differences while we maximize between-group differences. We defined notions of dissimilarity within the cluster and dissimilarity between clusters using the distance function "*d*" [40,41].
- From a statistical point of view, the methods correspond to a parametric approach. We assume that the unknown density, $p(x)$, of the data is a mixture of *k* densities, *pi(x)*, each of which corresponds to one of the *k* groups in the data. We assume that *pi(x)* comes from some parametric family (for example, Gaussian distributions) with unknown parameters, which we then estimate from the data [40,42].

Thus, density-based clustering takes a nonparametric approach, where the clusters in the data are high-density areas of density, $p(x)$. Density-based clustering methods do not require the number of clusters as input parameters, nor do they make any assumptions about the underlying density, $p(x)$, or the within-cluster variance that may exist in the data. Consequently, density-based clusters are not necessarily groups of points with high similarity within the cluster, as measured by the distance function *d*, but may have an "arbitrary shape" in feature space; sometimes, we also call them "natural pools" [40].

Likewise, we can evaluate the data density by analyzing the neighborhood of each data object. There are two possible ways to define the neighborhood of an object. First, when we express the neighborhood radius of an object as the Euclidean distance to the *k*-nearest neighbor, we define the neighborhood size dynamically depending on the data density. Object neighborhoods are relatively small in dense regions of the data space and considerably larger in less dense regions of the data space. The second option is to assume the same neighborhood radius for all data objects while pooling the data [43]. The density-based clustering algorithms that we used in the investigation are described in the following sections.

2.2.1. DBSCAN

Application density-based spatial clustering with noise (DBSCAN) is a density-based clustering algorithm proposed in [44], and we used it to evaluate the density of the data in a neighborhood of a predefined radius for each object and expressed it as the number of objects in that neighborhood. Therefore, we could identify three types of data objects in the DBSCAN pool: core objects, border objects, and peripheral objects [45].

- Core objects contains a predefined number of objects, *k*, in its neighborhood of radius *r*.
- We call the border objects if there are less than *k* objects in its neighborhood of radius *r*, but at least one of them is a core object.
- Peripheral objects is the object with less than *k* objects in its neighborhood of radius *r*, and none of them are a core object.

2.2.2. K-Means

K-Means clustering, or also known as the Lloyd–Forgy algorithm, is an unsupervised learning clustering algorithm first introduced in [46]. Its main objective is the classification of unlabeled data, based on characteristics; then, K-Means minimizes the intra-cluster variance and maximizes the inter-cluster variance, where each datum must be as close as possible to its group and as far as possible from another type of group [47,48]. Likewise,

we must consider that for K-Means to find the optimal number of clusters, it is possible to apply certain techniques, the most popular being the elbow method [49,50].

### 2.2.3. HDBSCAN

The density-based clustering algorithm based on hierarchical density estimates (HDBSCAN) is the proposal of the authors of [51], who generated an advanced DBSCAN method, improving the theoretical and practical aspects of the algorithm. The execution of the algorithm in five stages according to [52] is:

- Space transformation (stage 1)

We defined a new distance metric between points called "mutual reach distance" as:

$$d_{mreach-k}(a,\ b) = \max\{core_k(a),\ core_k(b),\ d(a,\ b)\} \tag{1}$$

Under this metric, dense points (with a low center distance) stay the same distance from each other, but sparser points move away to be at least their center distance from every other point.

- Construction of the minimum spanning tree (stage 2)

We started by considering the data as a weighted graph with the data points as vertices and an edge between two points with a weight equal to the mutual reach distance of those points. We considered a threshold value, starting high and steadily going down. We released any weighted edges above that threshold. As we released the edges, the graph in connected components started to become disconnected. Eventually, we will obtain a hierarchy of connected components (from fully connected to fully disconnected) at different threshold levels.

- Construction of a cluster hierarchy (stage 3)

We sorted the edges of the tree by distance (in increasing order) and then iterated, creating a new merge group for each edge.

- Condensation of the cluster hierarchy (stage 4)

Via the HDBSCAN parameter 'min_samples', we obtained the value for the minimum cluster size, then we traversed the hierarchy and, at each split, noted if one of the new clusters created by the split had fewer points than the minimum cluster size.

If it was the case that we obtained fewer points than the minimum size, we declared them as points that fall outside of a group. We resolved that the largest group retained the identity of the main group, marking which points fell out of the group and at what distance value that happened.

If, on the other hand, the split was into two groups (each at least as large as the minimum size), then we considered that we were dealing with a true split of the group and kept that split persistent in the tree. After traversing the entire hierarchy and doing this, we obtained a much smaller tree with a small number of nodes, each of which had data on how the size of the cluster at that node decreased over distance.

- Extraction of stable clusters from the condensed tree (stage 5)

Doing so involves calculating the stability of each previously formed group as follows:

$$\sum_{p\ \in\ cluster}\left(\lambda_p - \lambda_{birth}\right) \tag{2}$$

where lambda $\lambda = \frac{1}{distance}$, where $\lambda_p$ is the lambda value at which point $p$ "got out of the group", which is a value somewhere between $\lambda_{birth}$ (lambda value when the group broke up and became its own group) and $\lambda_{death}$ (lambda value when the group was split into smaller groups).

We declared all leaf nodes as selected clusters. Therefore, we proceeded through the tree (in reverse topological sort order). If the sum of the stabilities of the secondary clusters

was greater than the stability of the cluster, we set the stability of the cluster to be the sum of the secondary stabilities. If, on the other hand, the stability of the cluster was greater than the sum of its children, then we declared the cluster to be a selected cluster and deselected all its descendants. Once we reached the root node, we named it as current set of clusters.

### 2.3. Cluster Validation Techniques

Clustering methods have the objective of discovering characteristic groups present in a universe of data. In general, they tend to look for clusters whose members are close together (i.e., have a high degree of similarity) and are well-separated from other clusters [53]. Therefore, one of the most important problems in the field of cluster analysis is the validation of the results to find the number of groups or clusters best-suited to the data provided. For this, there are three approaches to verify the validity of the clusters: external, internal, and relative [54].

For the case of the study, given its nature and applied methodology, we used three internal validation indices:

- Silhouette coefficient

The Silhouette coefficient evaluates the validity of the clustering and selects the appropriate number of clusters. When the value of the coefficient is one or close to one, it indicates the good cohesion relationship between the elements of the cluster (internal) and the separability between the clusters (external). If the coefficient is zero or close to zero, it indicates that the clusters tend to overlap each other, and for values equal to or close to minus one, it indicates that the assignment to the cluster is incorrect, because the different clusters have greater similarity [55]:

$$s(i) = \frac{(b(i) - a(i))}{Max\{a(i), b(i)\}} \tag{3}$$

where $a(i)$ is the average distance within the cluster and $b(i)$ is the average distance of the nearest cluster for each sample [55].

- Calinski–Harabasz coefficient (*CH*)

The Calinski–Harabasz coefficient is the ratio of the sum of the inter-cluster spread to the within-cluster spread for all clusters (where we define the spread as the sum of the squared distances). For the case of this coefficient, we relate a higher score to a model with better-defined clusters [56]:

$$CH = \frac{trace(S_B)}{trace(S_w)} \cdot \frac{n_p - 1}{n_p - k} \tag{4}$$

where $(S_B)$ is the intergroup dispersion matrix, $(S_w)$ is the internal dispersion matrix, $n_p$ is the number of grouped samples, and $k$ is the number of clusters [57].

- Davies–Bouldin coefficient (*DB*)

This coefficient indicates the average "similarity" between clusters, where similarity is a measure that compares the distance between clusters to the size of the clusters themselves. Zero is the lowest possible score. Values closer to zero indicate better partitioning [58].

$$DB = \frac{1}{c} \sum_{i=1}^{c} Max_{i \neq j} \left\{ \frac{d(X_i) + d(X_j)}{d(c_i, c_j)} \right\} \tag{5}$$

where $c$ denotes the number of clusters, $i$ and $j$ are labeled clusters, so $d(X_i)$ and $d(X_j)$ are all samples in clusters $i$ and $j$ to their respective cluster centroids, and $d(c_i, c_j)$ is the distance between these centroids [57].

We also used three external validation indices:

- *F*-measure

The *F*-measure combines the precision and recall concepts. Precision is the ratio of the number of true positives to the number of false positives and is intuitively the ability of the classifier not to label a sample that is negative as positive. Recall is the ratio of the number of true positives to the number of false negatives [59].

$$Recall(i,j) = \frac{n_{ij}}{n_i} \tag{6}$$

$$Precision(i,j) = \frac{n_{ij}}{n_j} \tag{7}$$

where $n_{ij}$ is the number of elements of class $i$ that are in cluster $j$, $n_j$ is the number of elements in cluster $j$, and $n_i$ is the number of elements in class $i$. We calculated the *F*-measure of cluster $j$ and class $i$ with [57]:

$$F(i,j) = \frac{2 Recall(i,j) Precision(i,j)}{Precision(i,j) + Recall(i,j)} \tag{8}$$

The values of (8) are within the interval [0–1], and larger values indicate better quality of the grouping.

- *Purity*

*Purity* is the analysis of the clusters that yields the percentage value of the total number of elements that we correctly classified in the range of [0–1] [60]. For each cluster, the purity, $P_j = \frac{1}{n_j} Max_i(n_j^i)$, is the number of elements in $j$ with class label $i$. Then, $P_j$ represents a fraction of the total size of the cluster that the largest class of elements allocated. We obtained the total purity estimate from (9) [57]:

$$Purity = \sum_{j=1}^{m} \frac{n_j}{n} P_j \tag{9}$$

where $n_j$ is the size of cluster $j$, $m$ is the number of clusters, and $n$ is the total number of elements.

- *V*-measure

We describe this measure as the harmonic mean between the measures of homogeneity and completeness [57]:

$$V = \frac{(1+\beta) * Homogeneity * Completness}{\beta * Homogeneity + Completness} \tag{10}$$

The result of this measurement varies in a range from 0 to 1, where 1 is the best value and 0 is the worst.

- Random Adjusted Rand Index

The Rand index computes a similarity measure between two clusters by considering all pairs of samples and counting the pairs that map to the same or different clusters in the predicted and true clusters [61]:

$$RI = \frac{a+b}{C_2^{n_{samples}}} \tag{11}$$

where:

- $a$: The number of times a pair of elements are in the same group for both the actual and predicted grouping.
- $b$: The number of times that a pair of elements are neither in the same group for the real grouping, nor in the predicted one.
- $C_2^{n_{samples}}$: Total number of possible pairs in the dataset.

We then "likelihood-adjusted" the raw "*RI*" score into the *ARI* score using the following scheme:

$$ARI = \frac{RI - Expected\_RI}{max(RI) - Expected\_RI} \tag{12}$$

The result provided by this coefficient varies in a range from −1 to 1, where −1 is the worst result, 0 is a random result, and 1 is a completely similar result.

## 3. Materials and Methods

### 3.1. Type, Level, and Design of the Investigation

We carried out applied descriptive-level research, where we carried out the collection of data from the observation and subsequent processing to obtain a solution. The design was non-experimental for technological development, we did not manipulate any variables and we only limited ourselves to the study and analysis of pre-existing data to develop a solution that could improve current techniques.

### 3.2. Population and Sample

The population was undergraduate students enrolled during the academic semester 2021-II of the National University of San Martín, Peru: 5575 individuals. We calculated the sample with the finite population formula, 95% confidence level, resulting in 670 students. To select the sample, we sent emails to the entire university community and selected the first 670 participants who provided their informed consent and completed the provided psychological evaluation questionnaires.

### 3.3. Proposed Model

We generated a model for grouping students, according to the level of dropout risk, based on their responses to psychological tests, in which we integrated data mining and machine learning to replace the conventional mechanisms dedicated to tutoring; in this way, we improved academic and emotional follow-up. Figure 1 illustrates the proposed model consisting of five stages.

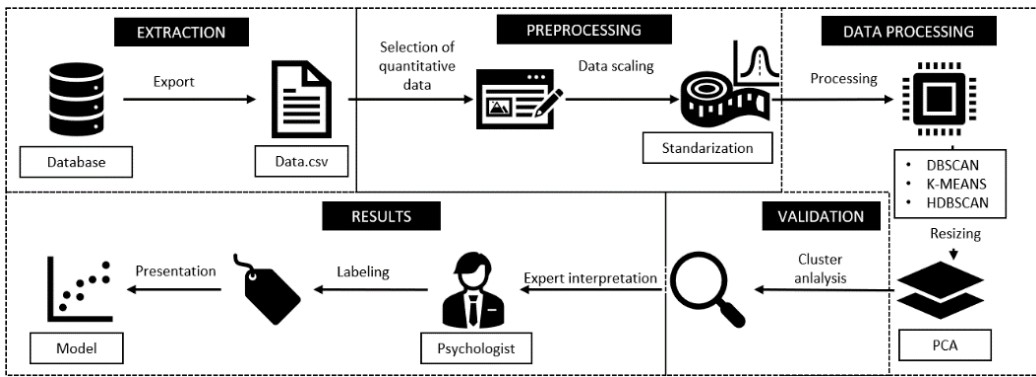

**Figure 1.** Proposed model.

### 3.4. Data Colection

We performed data collection through a web chatbot. The students responded to five psychological evaluation questionnaires validated and endorsed by previous studies, according to the following details:

- Study habits is a psychological questionnaire structured by 55 items. The questions evaluate the study habits and techniques used by students, which influence the learning process. The questionnaire is divided into five dimensions: how to organize to study, strategies used to solve tasks, methods used to prepare for an exam, the way you pay attention in class, and how do you study at home? The responses to the

questionnaire are dichotomous (always/never). The main objective of the instrument is to categorize the academic performance of students [62,63].

- Adaptation to university life is a questionnaire focused on evaluating the academic, institutional, and social dimensions of the students, with 50 structured items. It has Likert-type assessment scale responses, from the most negative rating to the most positive rating (totally disagree/ sometimes disagree/sometimes agree/totally agree). Specifically, the questionnaire helps to determine the nature of the adaptive process of the university student [64].
- Zung's Self-Assessment Depression Scale (SDS) is a standardized questionnaire that can be self-administered, based on norms elaborated in percentiles, with 20 structured items. It evaluates the affective, cognitive, and somatic aspects of the patients, through questions with a Likert-type assessment scale (never/sometimes/most of the time/always ). It has the aim of measuring the level of depression in a simple and specific way as a psychiatric disorder, allowing to categorize the depression level of an individual [65].
- The validated Spanish version of the Hamilton Anxiety Rating Scale (HARS) is a questionnaire and clinical assessment tool, structured in 14 items, with Likert-type responses (very disabling/severe/moderate/mild/none), which provide useful information about possible anxious-depressive symptoms to evaluate the symptomatology of an individual's level of anxiety [66].

For access to the chatbot web platform, we sent specific links to each student, periodically to their institutional email. We carried out this process during the academic semester 2021-II. Finally, we stored the data in a relational database.

### 3.5. Data Pre-Processing, Processing, and Visualization

During the investigation, we executed a set of data pre-processing, processing, and visualization techniques for further analysis. We stored the data in a digest schema database for simplicity, ease, and speed of processing. Then, we performed the data processing using an open-source integrated development environment (IDE) for scientific programming in Python, called Spyder V5.2.2.

With the purpose of executing the data processing through the DBSCAN, K-Means, and HDBSCAN unsupervised learning algorithms, we removed the data that did not contribute quantitative values to the model. To do this, we started by importing the data from the "Datos.csv" file to the "data" variable.

Once the data from the "data" variable file were imported, we eliminated the data from the "code" column, since they only had the function of identifying the student and did not provide relevant values for the model, as shown in Table 1.

**Table 1.** Data columns of the "data" variable.

| Column | Type |
|:---:|:---:|
| code | string |
| study habits | int |
| adaptation and coexistence | int |
| depression | int |
| anxiety | int |

Subsequently, we analyzed the descriptive statistics of the resulting set after the data selection and cleaning process, as shown in Table 2.

**Table 2.** Statistical data of the data columns in the "data" variable.

| Index | Study Habits | Adaptation and Coexistence | Depression | Anxiety |
|---|---|---|---|---|
| count | 670 | 670 | 670 | 670 |
| mean | 3.5731 | 0.6985 | 1.0940 | 0.2746 |
| SD | 0.9673 | 0.4784 | 0.3166 | 0.6975 |
| min | 1 | 0 | 0 | 0 |
| 25% | 3 | 0 | 1 | 0 |
| 50% | 4 | 1 | 1 | 0 |
| 75% | 4 | 1 | 1 | 0 |
| max | 5 | 3 | 3 | 3 |

From this, we identified heterogeneity of the data ranges (maximum and minimum) of the different columns. This is because the scores of the instruments, due to the number of items, are different, as we can see in Table 3.

**Table 3.** Scale of possible values in the data columns of the "data" variable.

| Column | Type | Labels |
|---|---|---|
| study habits | 0–5 | (very negative, negative, negative trend, positive trend, positive, very positive) |
| adaptation and coexistence | 0–2 | (low, medium, high) |
| depression | 0–3 | (normal, light, moderate, severe) |
| anxiety | 0–3 | (mild, moderate, serious, severe) |

Next, we proceeded to scale the data through normalization methods to provide the unsupervised DBSCAN, K-Means, and HDBSCAN algorithms with data in the same format and scale.

In Figure 2, we show the process of "standardization", where we scaled the data based on a normal distribution, adjusting the mean to 0 and the variance to 1.

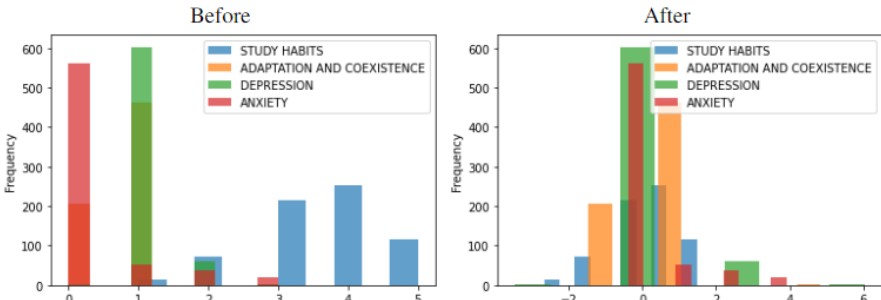

**Figure 2.** Contrast of data distributions before and after normalization.

- Processing with DBSCAN

We processed the data with the DBSCAN unsupervised learning algorithm. We presented the data in a multidimensional array, which we considered as a universal group. Considering the radius (*Eps*) for each of the points in a Euclidean space and through a minimum number of points (Min_pts), we defined the neighborhood of a point as:

$$N_{Eps}(p) = \{q \in D | dist(p, q) \leq Eps\} \tag{13}$$

Given the *Eps* and Min_pts parameters, DBSCAN randomly chooses a core point as a seed and retrieves all attainable density samples (within the *Eps* radius) from the seed to form a cluster, considering those points that do not belong to a cluster as noise.

To start the processing, it was crucial to have the parameters that the algorithm requires for its execution, *Eps* and MinPts. We calculated these parameters by iteratively executing the algorithm itself over a range of *Eps* and MinPts values to compile their results and contrast them with the project's objectives.

We based the method and criteria used for the selection of the algorithm parameters on the analysis of the coefficients: Silhouette coefficient, Calinski–Harabasz coefficient, and Davies–Bouldin coefficient.

We started by importing the necessary resources: numpy for numerical calculations, pandas for data manipulation in schemas called dataframes, and the "metrics" and "pre-processing" modules of the sci-kit learn library. We used product method of the itertools package to generate combinations based on the elements of two or more data lists. We executed the DBSCAN unsupervised learning algorithm based on the combinations of parameters generated by the product method.

Based on the objective of the investigation and with a range of *Eps* values from 0.2 to 2, we chose to assign an arbitrary Min_Pts range from 5 to 15. Then, with each of the proposed parameters, we generated the combinations based on the lists of generated data ranges and initialized the variables to store the data resulting from the iterative execution of the algorithm. We also applied a conditional filter to store results where the number of clusters was greater than 3, and less than 6.

From these results, we selected the parameters 1.7 and 6 for the *Eps* and Min_pts values, respectively, as shown in Table 4.

**Table 4.** Results obtained after DBSCAN execution.

| Index | Number of Clusters | Silhouette | Calinski–Harabasz | Davies–Bouldin | *Eps* | MinPts | Noise |
|---|---|---|---|---|---|---|---|
| 25 | 5 | 0.4972 | 190.7099 | 0.9571 | 1.7 | 6 | 9 |
| 35 | 4 | 0.4919 | 220.9307 | 1.1153 | 1.8 | 12 | 13 |
| 43 | 4 | 0.4919 | 220.9307 | 1.1153 | 1.9 | 12 | 13 |
| 51 | 4 | 0.4919 | 220.9307 | 1.1153 | 2 | 12 | 13 |

We established parameters according to Silhouette coefficient greater than 0, a high Calinski–Harabasz coefficient, a low Davies–Bouldin coefficient, a low number of noise-type points, and number of clusters greater than 3 and less than 6.

We applied DBSCAN to the obtained parameters. In the variable "labels", we stored the result of the computation of the algorithm on the dataset, resulting in a $670 \times 1$ list whose only column contains the labels of the clusters generated with its index in Y, the index corresponding to the student within the initial dataset.

From this result, we extracted the number of clusters = 5, the number of noise points = 9, the Silhouette coefficient = 0.4972, the Calinski–Harabasz coefficient =190.7099, and the Davies–Bouldin coefficient = 0.9571.

- Processing with K-Means

For data processing with the K-Means algorithm, we used the elbow method for the selection of the parameter "n_clusters". The elbow method consists of iteratively executing the clustering algorithm on a range of "n_clusters" that usually range from 1 to 10, and then, for each value of *k*, it calculates an average score for all the groups. We calculated the distortion score, which is the sum of the squared distances from each point to its assigned center.

When we plotted the values of these metrics, we could visually determine the best value for *k*. If the line graph looks like an arm, then the 'elbow' (the turning point in the

curve) is the best value of *k*. The 'arm' can be up or down, but if there is a strong inflection point, it is a good indication that the underlying model is a better fit for that point.

After obtaining the optimal value for the "n_clusters" parameter, shown in Figure 3, we applied the K-Means algorithm to the data.

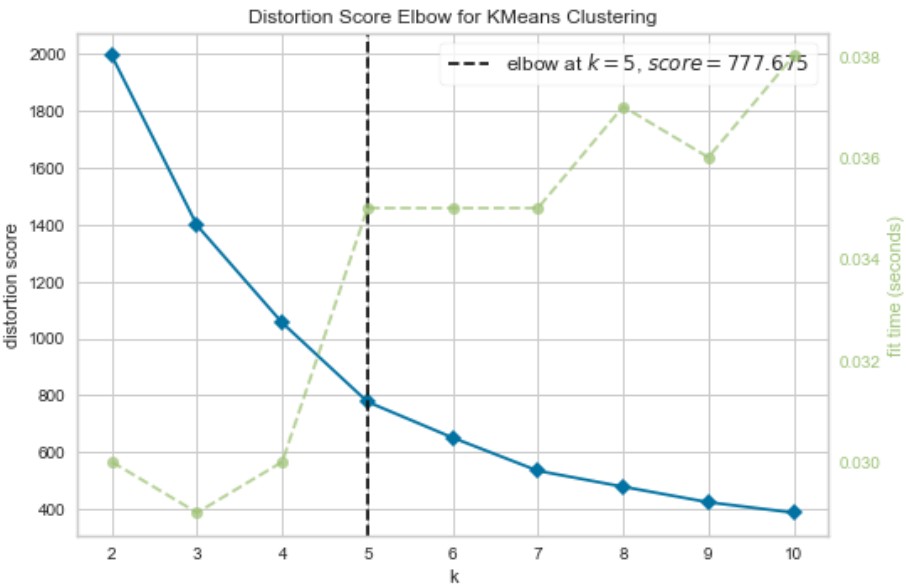

**Figure 3.** Result of the elbow method's application.

Based on the processing process, we obtained the following results: the number of clusters = 5, the Silhouette coefficient = 0.5586, the Calinski–Harabasz coefficient = 406.45089, and the Davies–Bouldin coefficient = 0.8001.

- Processing with HDBSCAN

In the case of HDBSCAN, it was possible to work with a single parameter: "min_cluster _size"; however, to have greater precision and control over the results of the coefficients, we decided to work with the second parameter "min_samples". Using a similar technique to the one we applied with DBSCAN, we based the selection method of the algorithm parameters on the analysis of the following coefficients: Silhouette coefficient, Calinski–Harabasz coefficient, and Davies–Bouldin coefficient.

We started by importing the necessary resources: numpy for number calculations, pandas for data manipulation in schemas called dataframes, the "metrics" and "preprocessing" modules of the sklearn library, and the product method of the itertools package to generate combinations based on the elements of two or more data lists. Then, we ran the HDBSCAN algorithm based on the combinations of parameters generated by the product method.

The range of values in the "min_cluster_size" parameter was from 15 to 80, and the range of values for the "min_samples" parameter was from 10 to 30. Next, with each of the proposed parameters, we generated the combinations based on to the lists of proposed data ranges and we initialized the variables to store the data resulting from the iterative execution of the algorithm. In addition, we applied a conditional filter to store results where the number of clusters was greater than 3, and less than 6, as well as a filter to avoid results that exceeded an amount of noise greater than 10% of the data.

Finally, after the execution of all cases of interest, we obtained 12 results. After this, we selected the parameters 55 and 19 as the values of "min_cluster_size" and "min_samples", respectively, as shown in Table 5.

As in the case of DBSCAN, we focused the parameter selection criteria on the analysis of internal validation indices: Silhouette coefficient with values greater than 0 and close to 1, the most ideal, a high Calinski–Harabasz coefficient, a low Davies–Bouldin coefficient,

with values close to 0 as the ideal, a number of noise-like points, and number of clusters greater than 3 and less than 6.

**Table 5.** Results after executing HDBSCAN.

| Index | Number of Clusters | Silhouette | Calinski–Harabasz | Davies–Bouldin | Minimum Cluster Size | Minimum Samples | Noise |
|-------|--------------------|------------|-------------------|----------------|----------------------|-----------------|-------|
| 8 | 5 | 0.6823 | 369.6459 | 0.6563 | 55 | 19 | 63 |
| 7 | 5 | 0.6704 | 349.5316 | 0.6677 | 55 | 18 | 59 |
| 2 | 5 | 0.6639 | 334.9714 | 0.6861 | 60 | 17 | 56 |
| 6 | 5 | 0.6639 | 334.9714 | 0.6861 | 60 | 17 | 56 |

We selected the number of clusters = 5, the number of noise points = 63, the Silhouette coefficient = 0.6823, the Calinski–Harabasz coefficient = 369.6459, and the Davies–Bouldin coefficient = 0.6563.

## 4. Analysis of Results and Discussion

When executing the algorithm on the proposed dataset, we obtained sets of results to which we applied two validation techniques (visual and internal) to verify the accuracy of the proposed model.

We performed the three-dimensional composition of the point cloud and its respective clusters for the graphic display of the results in each of the generated models; however, we generated the cloud in four dimensions, so it is impossible to represent it on a three-dimensional plane. For this reason, we resized the dataset using a data compression technique known as principal component analysis (PCA). For a graphical representation, we imported the PCA method from the "decomposition" module, which reduce the four-dimensional dataset to a three-dimensional dataset.

### 4.1. Visual Validation

We proceeded with the graphical representation of the resized dataset.

We could differentiate strongly defined structures after obtaining the graphical representation of the point cloud of the dataset and its respective clusters identified by colors in Figures 4–6. This three-dimensional representation is the result of resizing the four-dimensional dataset by applying PCA. However, despite not having the representation of the point cloud in its natural state, the feature reduction technique offers us a structure in which it is possible to visually recognize the clusters, being close to those elements of its own cluster and distant to those who do not belong or are alien to their group.

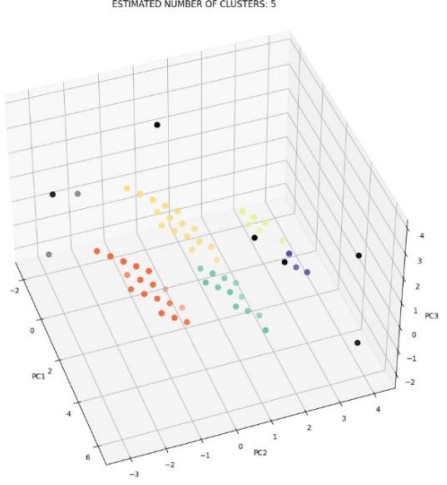

**Figure 4.** Visual representation of the three-dimensional clustered point cloud (DBSCAN).

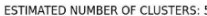

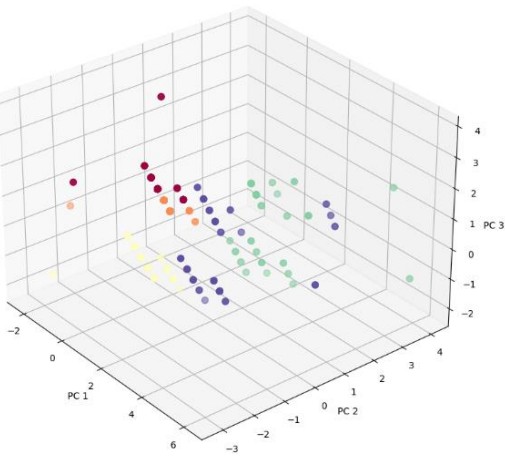

**Figure 5.** Visual representation of the clustered point cloud in three dimensions (K-Means).

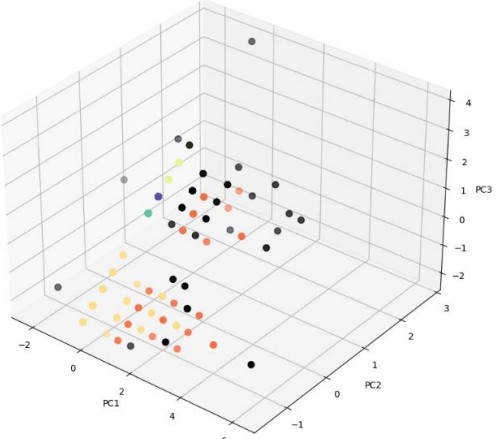

**Figure 6.** Visual representation of the three-dimensional clustered point cloud (HDBSCAN).

*4.2. Internal Validation*

Due to the lack of a previous classification of the sample used with which we can compare the external indices of the models generated by each algorithm, we considered it correct and prudent to make a comparison of the resulting internal indices: the Silhouette coefficient, the Calinski–Harabasz coefficient, and the Davies–Bouldin coefficient. Based on the results of the indices, we selected the best cases of each algorithm, as shown in Table 6.

**Table 6.** Comparative table of the best internal metrics resulting from the models.

| Algorithm | Silhouette | Calinski–Harabasz | Davies–Bouldin | Number of Clusters | Noise |
| --- | --- | --- | --- | --- | --- |
| DBSCAN | 0.4972 | 190.7099 | 0.9571 | 5 | 9 |
| K-Means | 0.5586 | 406.4509 | 0.8001 | 5 | - |
| HDBSCAN | 0.6823 | 369.6459 | 0.6563 | 5 | 63 |

From the results shown in Table 6, we found that the model generated with HDBSCAN was superior to the model generated with K-Means, and widely superior to the one generated with DBSCAN. This is because HDBSCAN had a better level of validity in two (Silhouette coefficient and Davies–Bouldin coefficient) of the three internal validation indices evaluated. However, for the third case, the Calinski–Harabasz coefficient generated

with the K-Means model was higher. This is because to calculate the Calinski–Harabasz coefficient, we used the centroids of each cluster as parameters. Thus, we obtained a much higher score in convex-shaped clusters and greater affinity with the shape of the clusters formed by the K-Means algorithm, that tended to have an almost spherical convex shape [47,48].

Based on the validation results of the internal HDBSCAN indices, we decided to categorize university students into five clusters, according to their risk of dropping out.

### 4.3. Expert Validacion

In collaboration with a team of three mental health experts, we interpreted and identified the recognized patterns in each data cluster. In Figure 7, the patterns in the distribution of the results after the generation of the clusters are presented.

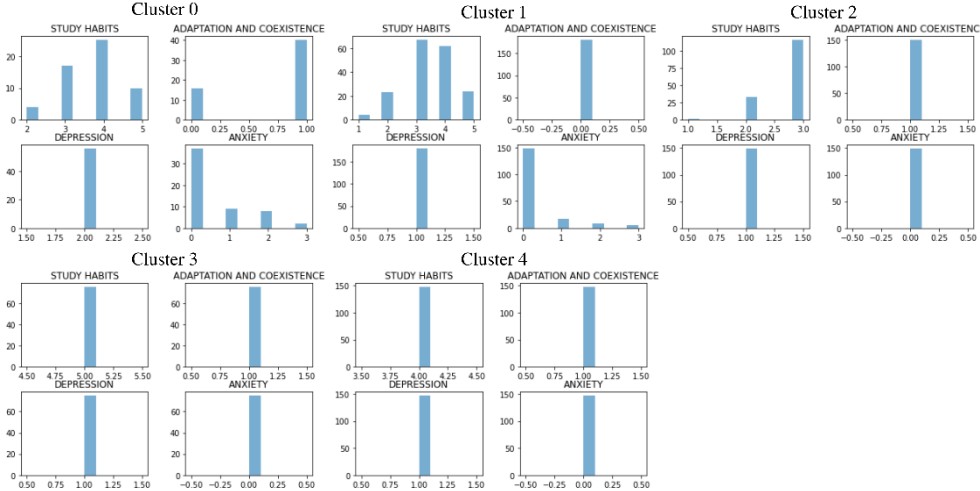

**Figure 7.** Distribution of results in each test with the members of the clusters 0, 1, 2, 3, and 4.

We proceeded with the labeling with the mean of the results in the different distributions and the criteria of the mental health professionals, based on the theories of the psychological questionnaires used [62–66].

From Table 7, the groupings of the sample of 670 students resulted in: very low dropout risk level = 75 students, low level = 147, medium level = 149, high level = 180, and very high level = 56, and 63 students belonged to the noise sector.

**Table 7.** Values of the mean of the data distribution according to clusters.

| Cluster | Study Habits | Adaptation and Coexistence | Depression | Anxiety | Risk Level |
|---------|--------------|----------------------------|------------|---------|------------|
| Cluster 0 | 3.7321 | 0.71429 | 2 | 0.5536 | 5 = Very high |
| Cluster 1 | 3.4389 | 0 | 1 | 0.2944 | 4 = High |
| Cluster 2 | 2.7651 | 1 | 1 | 0 | 3 = Middle |
| Cluster 3 | 5 | 1 | 1 | 0 | 1 = Very low |
| Cluster 4 | 4 | 1 | 1 | 0 | 2 = Low |

After the labeling process, we asked the team of experts to classify the students by evaluating the results of the applied psychological evaluation instruments. Then, we proceeded with the validation of external indices between the results of HDBSCAN and those of the experts to compare the accuracy of the model.

After we ran the validation of external indexes, we obtained Table 8 as a result.

**Table 8.** Values obtained after validation of external indexes.

| Index | Score |
|---|---|
| *F*-measure | 0.909 |
| *Purity* | 0.945 |
| *V*-measure | 0.869 |
| Adjusted Rand Index | 0.865 |

According to Table 8, the results obtained after comparing the classification performed through HDBSCAN with the classification provided by mental health professionals indicate a high level of precision in the *F*-measure (90.9%), which was consistent with the high similarity between the clusters calculated and the one predicted by the model with HDBSCAN (*Purity*: 94.5%, *V*-Measure: 86.9%, ARI: 86.5%).

The categorization of students provided a better picture of their dropout risk, which, associated with an early diagnosis, allows us to take corrective measures [34,35,67]. We must highlight that the use of data mining in conjunction with machine learning as tools allowed us to develop the main axis of the proposed model [15,18,21,31].

## 5. Conclusions

We developed a clustering model that integrates methodologies and data analysis and processing techniques, widely studied in the field of ICT, specifically in the field of unsupervised machine learning. This allowed us to obtain the successful categorization of undergraduate students from a Peruvian university into five levels based on the risk of desertion. HDBSCAN was the method that turned out to be the best option for data processing, as evidenced by the results of the internal validation indexes used to compare them with the K-Means and DBSCAN methods.

The resulting model serves as the basis of knowledge about the current view of university students. It can be replicated in other contexts, and it can be adjusted to other types of tests. For this, it would be necessary to standardize the input data types to generate values in less disperse ranges, to group them optimally. Likewise, it is scalable if we articulate joint efforts between mental health professionals and unsupervised learning techniques to generate a comprehensive solution that encompasses more dimensions of the psychological field. With this research, we contribute to the identification, prevention, and correction of various situations of psycho-emotional risk that university students may face.

**Author Contributions:** The manuscript was conceptualized by M.A.V.-C.; methodology, M.A.V.-C. and J.J.-D.; software, L.S.-R. and R.I.; validation, J.R.N.-C. and L.P.; formal analysis, P.V.-R.; data curation, L.S.-R. and E.A.H.-T.; writing—proofreading and editing, M.A.V.-C. and J.R.N.-C. All authors have read and agreed to the published version of the manuscript.

**Funding:** Thanks to the Universidad Nacional de San Martín for the financing of the project "Caracterización del proceso de tutoría a estudiantes de la UNSM aplicando un modelo de atención virtual basado en chatbots", financed by Resolution No. 359-2021-UNSM/CU-R.

**Institutional Review Board Statement:** Not applicable.

**Informed Consent Statement:** We obtained informed consent from all subjects involved in the study.

**Data Availability Statement:** The data of the survey carried out on students are available upon request at: The Research Unit of the Universidad Nacional de San Martín.

**Conflicts of Interest:** The authors declare no conflict of interest.

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
