# Peer review of "Density-Based Unsupervised Learning Algorithm to Categorize College Students into Dropout Risk Levels"

_data_

Round 1
Reviewer 1 Report (Previous Reviewer 4)
I enjoyed reading this manuscript. It tackles a fundamental problem for many universities.
The provided model which is based on machine learning offers an early diagnostic tool that can be used in other universities to monitor students at risk. It would be very useful if the model is applied with different data from different universities.
Author Response
Consulte el archivo adjunto.

Reviewer 2 Report (Previous Reviewer 2)
Overall, I don't see enough improvements over the previous version. The authors have more details on parameter selection and evaluation metrics. However, there are still a lot of issues need to be addressed in the paper.
In the abstract, the authors report the clustering scores in 8 decimal places. In the abstract and in the paper, the authors should report results in 8 decimal places. This is too long. 3 or 4 decimal places are enough.
The authors should clearly mention their contribution at the end of the introduction and a description/organization of the paper at the end of the introduction.
Section 3.2, the authors should explain more about the sampling process. How the 670 students were selected? Any limitations or representation bias in the sampling processing?
Section 3.4, line 287-292. The authors mention "NLP model". What is it? I don't see any NLP model in the questionnaire. Also in Section 3.4, the authors should explain more on the questionnaires. What questions are asked in the questionnaires. The authors could provide more details in the appendix.
Section 3.5, the authors normalized the feature values using the standardscaler. I don't think this is the best scalar here because the features are categorical. Also, what do the feature values mean in the categorical features? For example, there are 5 different feature values for study habits (1,2,3,4,5). What does each value mean? The authors should clarify the feature values and the training data.
In section 3.5, in the parameter selection process, the authors did it on the whole training data on 670 samples and determine the best set of tuning parameters. However, this approach can lead to overfitting. In machine learning, a better approach to do it would split the whole data into three parts: training set, validation set and test set with about 60%-20%-20% distribution. To select the best tuning parameter values, the authors can iteratively train the algorithm over a range of tuning parameter values on the training set and apply the model on the validation set to get noise and Silhouette scores and choose the set of the tuning parameters based on results from validation set. After the best parameter values are chosen, the authors can validate each model on the test set This approach can avoid overfilling and show the robustness of the machine learning model.
There are many figures and tables never mentioned in the paper. For example Table 4-5 and Figure 2-8, the authors have the tables/figures in the paper, but never talk about them. By the way, Figure 7 is missing in the paper.
In Section 4.3, expert interpretation, I think the authors could change the methodology a little bit. For example, the authors can consult the experts to determine if there could be more or less clusters than 5 and what are the criteria/factors to determine dropout risk level. A even better validation is to compare the risk level to the university stage test results, but the test results are not available until the end of secondary education. Overall, the section name "expert interpretation" is misleading. The readers would expect some expert validation on clustering results.
Based on the experiment and results, the authors never compare their proposed approach to existing methods predicting dropout risk levels. I want to see if the proposed approach is actually better than existing tools, but the authors do not have such comparison in the paper.
Author Response
Please see the attachment

Reviewer 3 Report (Previous Reviewer 1)
To enrich your introduction and discussion, kindly consider more works on the application of machine learning in education as recommended below:
1. https://datascience.codata.org/articles/10.5334/dsj-2019-014/
2. https://www.mdpi.com/2076-3417/12/19/9467/htm
3. https://link.springer.com/article/10.1007/s43926-022-00023-0
Author Response
Please see the attachment

Reviewer 4 Report (New Reviewer)
I recommend accepting the paper
Round 2
Reviewer 2 Report (Previous Reviewer 2)
Thanks for the revision. I enjoyed reading this paper. I hope this application will help college students.
This manuscript is a resubmission of an earlier submission. The following is a list of the peer review reports and author responses from that submission.
Round 1
Reviewer 1 Report
The authors present the density-based unsupervised learning algorithm to categorize college students into dropout risk levels. The study is well presented and summary of the obtained results are briefly demonstrated in the abstract. I recommend the authors to correct the grammatical errors and include part of the future works in the last section (conclusions) if any.
Reviewer 2 Report
This paper presents an unsupervised clustering model using DBSCAN to replace the conventional mechanisms to predict dropout risk for university students. The model groups students into five levels of dropout risk and the authors use internal and external evaluation to their model.
The authors did a great job on explaining their methodology from data collection to parameter tuning in detail. However, there are several issues with the experimental design and presentation.
Experimental design:
- In section 3.5, the authors describe how they chose the best tuning parameter values for DBSCAN. In the parameter selection process, the authors did it on the whole training data on 670 samples and determine the best set of tuning parameters. However, this approach can lead to overfitting. In machine learning, a better approach to do it would split the whole data into three parts: training set, validation set and test set with about 60%-20%-20% distribution. To select the best tuning parameter values, the authors can iteratively train the algorithm over a range of Eps and MinPts values on the training set and apply the model on the validation set to get noise and Silhouette scores and choose the set of the tuning parameters based on results from validation set. After the best parameter values are chosen, the authors can validate the DBSCAN model on the test set This approach can avoid overfilling and show the robustness of the machine learning model.
- In Section 4.2, in the last paragraph, the authors mention Silhouette scores from two other models [36] and [37]. The authors need to explain the model. In addition, reference 36 and 37 were missing in the reference.
- In Section 4.3, validation by expert, I think the authors could change the methodology a little bit. For example, the authors can consult the experts to determine if there could be more or less clusters than 5 and what are the criteria/factors to determine dropout risk level. A even better validation is to compare the risk level to the university stage test results, but the test results are not available until the end of secondary education.
- I suggest the authors to use another clustering evaluation metric such as Calinski-Harabasz index in internal evaluation.
- In section 2.2, the authors mention why they did not use other clustering algorithms. However, for validation, I think the authors still need to include some other clustering methods such as K mean with k=5. Showing the results from other clustering algorithms can empirically show that DBSCAN is the best in this scenario. Results from other clustering algorithms can be added into internal validation
Presentation/organization
- In the abstract, the authors use the abbreviation “ICTs” without showing the full name. The authors should add the full name before using the abbreviation.
- In section 2.1, the authors describe previous work on predicting university dropouts and mention the model accuracy. The authors could describe more about what kind of models they use.
- in section 2.1, last paragraph, the authors mention problems with previous models. The author could have some of the problems in the introduction to show the novelty and contributions.
- The authors should convert screenshots in Figure 3 and Figure 6 into tables
- Font size in tables e.g. Table 1, 2 and 3 is much large than the font size in the text.
- Reference is incomplete. It has 38 references in the text, but there are only 34 references in the references section.
Reviewer 3 Report
Overall, the manuscript needs to address the “so what” question. The manuscript needs to articulate the implications for research and practice.
Introduction: It needs to be expanded with evidence from the literature on how higher education institutions conduct student assessments (emotional, aptitudinal, and student attrition).
The problem and purpose need to be expanded and provide the reasoning behind the proposed methodology. Authors need to justify the selection of DBSCAN over HBDSCAN, which is a more robust option.
I expected the contributions and a general description of the manuscript’s organization at the end of the introduction.
Theoretical foundations: Under the university dropout section, only five references were indicated, and a few of them are old. I encourage the authors to do a comprehensive literature review on student attrition models in higher education. Instead of citing specific studies, it would be helpful to mention the attributes, algorithms, and types of tasks (supervised, unsupervised, etc.) used across the literature.
The clustering (2.2) and clustering based on density (2.3) sections can be integrated into one section. It must articulate parametric and nonparametric methods' differences, strengths, and weaknesses.
In the DBSCAN section (2.4), readers will benefit from understanding the available clustering algorithms (SVM, K-Means, Affinity Propagation, HBDSCAN, etc) and highlighting the origins of DBSCAN by Ester et al. (1996) and its strengths/challenges over other clustering techniques.
The cluster validation section (2.5) needs to expand on validation metrics for DBSCAN. I encourage the authors to provide a complete discussion on parameters (epsilon, minimum samples, and mean silhouette coefficient). Mathematical equations are useful for an expert audience. Please consider other ways to articulate these terms to target different audiences.
Methods: The study needs to highlight the mixed-methods nature of the study as an overview of this section.
In section 3.2, the sample size of 670 needs to be broken down by school. The authors need to articulate the purpose of the random sampling and whether or not each school is represented. Also, the authors need to state the limitations of the sampling procedure.
In section 3.4, figure 2 is unnecessary and does not provide information about the survey items. Instead, it would be more beneficial to articulate how students took the survey using the chatbot. Replace figure 2 with an appendix to encapsulate a sample or all survey items.
In section 3.5, table 2 seems redundant since the Likert scale is already described in section 3.4 and figure 3. Also, please kindly disclose the standardization packages and functions involved (preprocessing.scale, standardscaler?).
In section 4.3, please disclose the criteria and how the experts validated the clusters. I expected an interpretation of the clusters presented in table 3 or what it means to be high or low risk. Was there an inter-rater reliability measure among experts?
After the analysis section, a new discussion section needs to be separated to state the implications for research and practice. This new section would address the “so what” question.
In the conclusion section, there should be a brief recap of the study and the future directions or implementations, if any.
Reviewer 4 Report
Line 10, you use an acronyms (ICT) in the abstract.
The introduction is poorly written, it says nothing about the proposed method, the survey, or the proposed solution. I strongly recommend revising the introduction.
The organization of the article is not mentioned in the introduction. Usually, the last paragraph of the introduction talks about how the article is put together.
2.1 claims that current methods have issues with scalability, replication, and bias. Those issues were never addressed by the proposed methodology, meaning that, we did not know if the proposed model was scalable, replicable, or non-biased.
2.2 (and parts of 2.3 and 2.4 ) were copied word for word from other references without using quotation marks.
Density-based clustering and DBSCAN were introduced and suggested without justification or even an intuitive explanation of why they could be efficient in this particular task (drop out prediction).
Sub-section 3.1 is vague, and I do not know what purpose it serves. Line 145 in this sub-section says: "Applied descriptive level research, where we carry out data collection from observation and subsequent processing to obtain a solution" This is an odd way to start a paragraph. Did you mean "we carried out applied descriptive... etc."?
Sub-section 3.4 should be extended, each bullet point/questionnaire should be explained further, and for example, what do you mean by 55 items? Where did they come from?
Questionnaires and sample answers should be added (possibly as an annex) to the article.
Lines from 179 to 182: this is interesting. Why don’t you talk more about this NLP model? Readers might be interested in this.
Lines from 197 to 200: the process of removing these data items is vague. What did you remove from the dataset? What is the specific criteria that you followed?
Many technical details can be skipped, especially from line 241 to 257.
Human judgment evaluation must be carried out where human judges must manually cluster a sample of the data, and a correlation between manual clustering and your approach must be calculated. Then the Kappa coefficient must be calculated to measure inter-judges agreement (and reliability).
The results were not thoroughly discussed.
No benchmarking with other methods and algorithms was presented.
No comparison between your results and SOTA was presented.
The conclusion is very brief. And it does show how this research is significant.
Again, related work was criticized in this manuscript because of the lack of scalability, being biased, and being non-replicable. However, this was not a criteria for assessment of the results. We cannot see if the proposed method is in fact replicable, scalable, and unbiased.